# Perinatal factors and hospitalisations for severe childhood infections: a population-based cohort study in Sweden

Samuel Videholm [1], Urban Kostenniemi,[1,2] Torbjörn Lind,[1]
Sven-Arne Silfverdal [1]

¹Department of Clinical Sciences, Pediatrics, Umeå University, Umeå, Sweden
²Department of Clinical Microbiology, Infectious Diseases, Umeå University, Umeå, Sweden

**Correspondence to**
Dr Samuel Videholm;
samuel.videholm@umu.se

## ABSTRACT

**Objective** To examine the association between perinatal factors and hospitalisations for sepsis and bacterial meningitis in early childhood (from 28 days to 2 years of age).

**Design** A population-based cohort study. The Swedish Medical Birth Register was combined with the National Inpatient Register, the Cause of Death Register, the Total Population Register and the Longitudinal integration database for health insurance and labour market studies. Associations between perinatal factors and hospitalisations were examined using negative binomial regression models.

**Setting** Sweden.

**Participants** 1 406 547 children born in Sweden between 1997 and 2013.

**Main outcome measures** Hospital admissions for sepsis and bacterial meningitis recorded between 28 days and 2 years of life.

**Results** Gestational age was inversely associated with severe infections, that is, extreme prematurity was strongly associated with an increased risk of sepsis, adjusted incidence rate ratio (aIRR) 10.37 (95% CI 6.78 to 15.86) and meningitis aIRR 6.22 (95% CI 2.28 to 16.94). The presence of congenital malformation was associated with sepsis aIRR 3.89 (95% CI 3.17 to 4.77) and meningitis aIRR 1.69 (95% CI 1.09 to 2.62). Moreover, children born small or large for gestational age were more likely to be hospitalised for sepsis and children exposed to maternal smoking were more likely to be hospitalised for meningitis.

**Conclusions** Prematurity and several other perinatal factors were associated with an increased risk of severe infections in young children. Therefore, clinical guidelines for risk assessment of infections in young children should consider perinatal factors.

## BACKGROUND

Sepsis and bacterial meningitis are major causes of morbidity and mortality in young children. After the neonatal period (from birth to 28 days), the global incidence of paediatric sepsis was estimated to be 1.2 million cases per year in 2013 and the global incidence of paediatric meningitis was

## STRENGTHS AND LIMITATIONS OF THIS STUDY

⇒ The large number of events allowed us to estimate incidence rates of sepsis and bacterial meningitis by age in months and year of birth.
⇒ The linkage of high-quality health and administrative registers enabled us to examine key birth, pregnancy and sociodemographic characteristics.
⇒ Hospitalisations were identified using deidentified hospital discharge data; infections could not be confirmed by laboratory findings.
⇒ Information on perinatal factors was missing for 17% of children; nevertheless, results from analyses with imputed data were overall consistent with results from analyses restricted to children with complete data.

around 280 000 cases per year in 2016.[1 2] In high-income countries, sepsis and bacterial meningitis still contribute significantly to the burden of childhood disease.[3–5]

Sepsis and bacterial meningitis are critical illnesses requiring early identification and rapid management.[6 7] Early identification is challenging; clinical guidelines recommend that clinicians use a combination of symptoms, vital signs and laboratory findings.[7 8] In the neonatal period, clinicians are also recommended to consider perinatal risk factors including gestational age, maternal intra-amniotic infection and maternal group B *Streptococcus* colonisation.[9–11] After the neonatal period, children with certain perinatal conditions, for example, low gestational age and low birth weight are known to be more susceptible to severe infections.[12 13] However, perinatal risk factors are not regularly considered in clinical guidelines for severe infections after the neonatal period.[7 8 11 14]

Therefore, we examined associations between perinatal factors, specifically birth, pregnancy and sociodemographic characteristics, and hospitalisations for sepsis and

bacterial meningitis in early childhood (from 28 days to 2 years of age).

## METHODS

### Patient and public involvement

This study used retrospectively collected register data. Therefore, patients or members of the public were not involved in the design or conduct of this study.

### Study population

We conducted a population-based cohort study including children born in Sweden between 1997 and 2013, who resided in Sweden at 28 days of age. National health and administrative registers were combined to a dataset using the national registration number, a unique personal identification number assigned to all Swedish residents. The dataset included information from the Medical Birth Register which covers over 98%–99% of all births and contains information on prenatal, delivery and neonatal care, the National Inpatient Register which holds information on 99% of all inpatient hospital admissions, the Longitudinal integration database for health, insurance and labour market studies which contains socioeconomic data, the Cause of Death Register which covers over 97% of all deaths and the Total Population Register.[15–18] The data were anonymised and linked by the Swedish National Board of Health and Welfare.

### Main exposures

Information on pregnancy and birth characteristics were obtained from the Medical Birth Register. Gestational age was estimated using fetal ultrasound measurements in the early second trimester (performed in around 95% of pregnancies) or information of the last menstrual period and categorised as extremely premature (22–27 weeks), very premature (28–31 weeks), moderate premature (32–36 weeks), term (37–41 weeks) and post-term (≥42 weeks).[19] Sex was categorised as male or female. Small for gestational age (SGA) was defined as a birth weight below the 10th percentile for gestational age at birth.[19] Large for gestational age (LGA) was defined as a birth weight over the 90th percentile for gestational age at birth. Congenital malformation included abnormalities detected during the first 28 days; these malformations were recorded using International Classification of Diseases, 10th revision (ICD-10) codes Q00–Q99.[20] Maternal age was defined as age at delivery. Maternal smoking during the pregnancy was self-reported (yes or no) during the first antenatal care visit, normally between 8 and 12 weeks of gestation.[21] Maternal body mass index (BMI) during the pregnancy was calculated from weight measured at the first antenatal care visit and self-reported height, and categorised as underweight ($<18.5\,\mathrm{kg/m^2}$), normal ($18.5–24.9\,\mathrm{kg/m^2}$), overweight ($25.0–29.9\,\mathrm{kg/m^2}$) and obese ($≥30.0\,\mathrm{kg/m^2}$).[15] Parity was defined as the number of live births including the index child.

Information on maternal education level was retrieved from the Longitudinal integration database for health, insurance and labour market studies and categorised as secondary school or less (≤9 years), upper secondary school (10–12 years), short postsecondary education (13–14 years) and long postsecondary education (≥15 years).

Information on maternal country of birth was obtained from the Total Population Register and categorised as Sweden, Other Nordic, Other Europe and North America, Asia, Africa and Other. The Total Population Register was also used to identify children who migrated and the Cause of Death Register was used to identify children who died.

### Outcomes

The two main outcomes were number of hospitalisations for sepsis and number of hospitalisations for bacterial meningitis. Information on hospital admissions with a principal diagnosis of sepsis (ICD-10 codes: A39.1-A39.9, A40, A41, A48.3, A02.1 and A32.7) and/or bacterial meningitis (ICD-10 codes: A39.0, A32.1 and G00) were retrieved from the National Inpatient Register. Only hospitalisations with admission dates between 28 days and 2 years of age were included. Consequently, infections with admission date in the neonatal period were not included. Readmissions within 30 days were excluded.

### Statistical analysis

We estimated incidence rates of sepsis and bacterial meningitis as the number of hospital admissions per 100 000 person-years (PY) at risk.

We examined associations between birth (sex, gestational age, SGA, LGA and congenital malformation), pregnancy (smoking during pregnancy, pregnancy BMI and parity) and sociodemographic (maternal age, maternal education level and maternal country of birth) characteristics, and hospitalisation for sepsis and bacterial meningitis using negative binomial regression models. Crude and adjusted analyses were performed for all variables. In the adjusted analyses, all analyses were adjusted for birth characteristics, pregnancy characteristics, sociodemographic characteristics and time trends (year of birth: 1997–2002, 2003–2008 and 2009–2013). All models were restricted to observations with complete data on all variables. Log follow-up time (in days) was used as offset. Clustering within families was accounted for using generalised estimating equations with robust standard errors.[22] Results were presented as adjusted incidence rate ratios (aIRRs) with 95% CIs.

In interaction analysis, we examined interactions between sex and prematurity (gestational age <37 weeks), SGA, LGA, congenital malformation and maternal smoking; between prematurity and SGA, LGA, congenital malformation and maternal smoking. Interactions were estimated on an additive scale by calculating the relative excess risk due to interaction (RERI). RERI was calculated using the following formula: $\mathrm{RERI} = \mathrm{aIRR}_{11}$

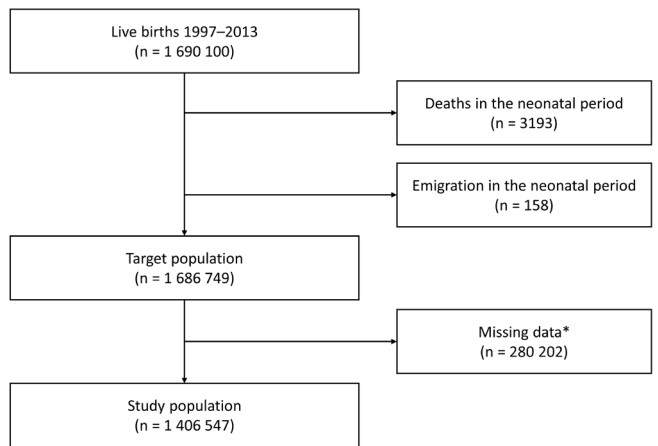

**Figure 1** Flow chart of the study population. Complete-case analyses included all children in the study population. Multiple imputation analyses included all children in the target population. *Children with missing data on any variable (sex, gestational age, small for gestational age, large for gestational age, congenital malformation, maternal age, smoking during pregnancy, pregnancy body mass index, parity, maternal education level, maternal country of birth and year of birth) were excluded.

$- \text{aIRR}_{10} - \text{aIRR}_{01} +1$; 95% CIs were estimated using the delta method.[23 24]

In sensitivity analysis, we imputed missing data using multiple imputation.[25] The predictive model included all variables in the substantive models and both outcomes (hospitalisation for sepsis or bacterial meningitis). Variables were imputed in order of increasing missingness using logistic regression models (categorical variables) and linear regression models (continuous variables). Ten imputed datasets were generated.

All statistical analyses were performed using Stata Statistical Software: Release V.14.

## RESULTS
### Study population
The Medical Birth Register included 1 690 100 live births between 1997 and 2013. Children who died (n=3193) or emigrated (n=158) in the neonatal period were removed, leaving 1 686 749 children who resided in Sweden at 28 days of age (the target population). We excluded children with missing data (n=280 202), leaving 1 406 547 children (figure 1). All children were followed until 2 years of age or censoring due to death (n=1417) or emigration (n=7185). In total, the study included 2 698 039 PY of follow-up time, 1298 hospitalisations for sepsis and 405 hospitalisations for bacterial meningitis.

Table 1 presents background characteristics of the study population. During the study period, 1011 children were hospitalised at least once for sepsis, 382 children were hospitalised at least once for bacterial meningitis and 18 children were hospitalised at least once for both sepsis and bacterial meningitis. Children hospitalised for severe infections were more likely to have certain characteristics,

for example, premature birth, SGA and low maternal education level.

### Incidence rates
Figure 2 shows the incidence rates of sepsis and bacterial meningitis hospitalisations by age in months. The incidence rate of sepsis decreased until the third month of life. Thereafter, it remained around 45 hospitalisations per 100 000 PY. The incidence rate of bacterial meningitis decreased until the fourth month of life, thereafter it was around 20 hospitalisations per 100 000 PY in the first year of life and around 10 hospitalisations per 100 000 PY in the second year of life.

Figure 3 shows the incidence rates of sepsis and bacterial meningitis hospitalisations by year of birth. The incidence rate of bacterial meningitis was around 20 hospitalisations per 100 000 PY in children born 1997–2008. Thereafter, the incidence rate of bacterial meningitis decreased markedly. This was mainly due to a lower incidence rate of pneumococcal meningitis. Online supplemental appendix A includes information on incidence rates for categories of bacterial meningitis grouped by year of birth.

### Perinatal factors
Table 2 presents associations between perinatal factors and hospitalisations for sepsis and bacterial meningitis. Gestational age was inversely associated with severe infections. In comparison with term children, children born extremely preterm were much more likely to be hospitalised for sepsis aIRR 10.37 (95% CI 6.78 to 15.86) and bacterial meningitis aIRR 6.22 (95% CI 2.28 to 16.94). Presence of congenital malformation was associated with both sepsis aIRR 3.89 (95% CI 3.17 to 4.77) and bacterial meningitis aIRR 1.69 (95% CI 1.09 to 2.62). An increased risk of sepsis was observed in children born SGA aIRR 2.44 (95% CI 1.83 to 3.25) or LGA aIRR 1.71 (95% CI 1.19 to 2.44). Children born to mothers who smoked during pregnancy were more likely to be hospitalised for bacterial meningitis aIRR 1.46 (95% CI 1.05 to 2.04). Low maternal education (≤9 years) was in comparison with post-secondary education (≥15 years) associated with both sepsis and bacterial meningitis in the unadjusted analyses. Sensitivity analysis showed similar estimates after multiple imputation, except for a stronger association between extreme prematurity and meningitis. Online supplemental appendix B includes results from multiple imputation models.

Table 3 presents results from interaction analyses. An additive interaction was found between prematurity (gestational age <37 weeks) and congenital malformations on the risk of sepsis RERI 7.75 (95% CI 3.45 to 12.06); no other additive interactions were observed.

## DISCUSSION
We found that prematurity, in a dose-dependent manner, and congenital malformations were associated with hospitalisations for sepsis and bacterial meningitis in early

**Table 1** Birth, pregnancy and sociodemographic characteristics of the study population

| | Included | | | Excluded* |
|---|---|---|---|---|
| | Sepsis | Bacterial meningitis | All included | |
| | n=1011 | n=382 | n=1 406 547 | n=280 202 |
| Sex | | | | |
| Male | 55.8 (564) | 57.9 (221) | 51.4 (722 988) | 51.4 (144 079) |
| Female | 44.2 (447) | 42.4 (162) | 48.6 (683 559) | 48.6 (136 117) |
| Gestation age† | | | | |
| Extremely preterm | 2.5 (25) | 1.0 (4) | 0.1 (2012) | 0.7 (1927) |
| Very preterm | 2.4 (24) | 2.1 (8) | 0.4 (5457) | 1.6 (4418) |
| Moderate preterm | 8.8 (89) | 5.8 (22) | 4.0 (56 291) | 10.3 (28 979) |
| Term | 80.7 (816) | 84.8 (324) | 88.3 (1 242 234) | 81.4 (227 950) |
| Post-term | 5.6 (57) | 6.5 (25) | 7.1 (100 553) | 5.7 (15 926) |
| SGA | | | | |
| Yes | 7.8 (79) | 3.7 (14) | 2.1 (30 099) | 2.1 (5972) |
| No | 92.2 (932) | 96.6 (369) | 97.9 (1 376 448) | 78.7 (220 473) |
| LGA | | | | |
| Yes | 5.1 (52) | 5.0 (19) | 3.7 (51 816) | 3.0 (8334) |
| No | 94.9 (959) | 95.3 (364) | 96.3 (1 354 731) | 77.8 (218 111) |
| Congenital malformation‡ | | | | |
| Yes | 14.1 (143) | 6.0 (23) | 3.4 (48 209) | 3.7 (10 401) |
| No | 85.9 (868) | 94.2 (360) | 96.6 (1 358 338) | 96.3 (269 801) |
| Maternal age | | | | |
| Mean (SD) | 30.0 (5.1) | 29.7 (5.5) | 30.1 (5.1) | 30.0 (5.5) |
| Maternal smoking | | | | |
| No smoking | 90.6 (916) | 86.9 (332) | 91.5 (1 286 708) | 61.3 (171 790) |
| Smoking | 9.4 (95) | 13.4 (51) | 8.5 (119 839) | 6.9 (19 472) |
| Pregnancy BMI§ | | | | |
| Underweight | 2.1 (21) | 2.4 (9) | 2.4 (33 836) | 1.0 (2895) |
| Normal | 59.8 (605) | 60.7 (232) | 61.6 (865 874) | 20.3 (56 838) |
| Overweight | 25.7 (260) | 25.1 (96) | 24.8 (348 695) | 9.1 (25 600) |
| Obese | 12.4 (125) | 12.0 (46) | 11.2 (158 142) | 4.3 (12 030) |
| Parity | | | | |
| Mean (SD) | 1.9 (1.0) | 1.8 (1.1) | 1.8 (1.0) | 2.0 (1.2) |
| Maternal education in years | | | | |
| ≤9 | 14.8 (150) | 15.4 (59) | 11.6 (162 814) | 10.6 (29 682) |
| 10–12 | 42.0 (425) | 42.4 (162) | 43.6 (613 370) | 36.6 (102 593) |
| 13–14 | 12.7 (127) | 16.0 (61) | 13.7 (192 659) | 12.6 (35 362) |
| ≥15 | 30.6 (309) | 26.4 (101) | 31.1 (437 704) | 25.7 (71 907) |
| Maternal country of birth | | | | |
| Sweden | 77.3 (782) | 80.6 (308) | 81.2 (1 142 682) | 70.9 (198 735) |
| Other Nordic | 1.8 (18) | 1.8 (7) | 1.7 (23 403) | 2.6 (7173) |
| Other Europe and North America | 8.8 (89) | 6.8 (26) | 6.2 (87 864) | 9.0 (25 278) |
| Asia | 8.6 (87) | 6.3 (24) | 7.4 (103 858) | 10.8 (30 392) |
| Africa | 2.4 (24) | 3.4 (13) | 2.4 (33 230) | 5.2 (14 644) |
| Other | 1.1 (11) | 1.3 (5) | 1.1 (15 510) | 1.3 (3688) |
| Year of birth | | | | |
| 1997–2002 | 27.1 (274) | 33.0 (126) | 29.5 (41 4783) | 40.3 (112 908) |
| 2003–2008 | 40.6 (410) | 46.1 (176) | 35.8 (503 841) | 37.1 (103 988) |

Continued

**Table 1** Continued

| | Included | | | Excluded* |
|---|---|---|---|---|
| | Sepsis | Bacterial meningitis | All included | |
| | n=1011 | n=382 | n=1 406 547 | n=280 202 |
| 2009–2013 | 32.3 (327) | 21.2 (81) | 34.7 (487 923) | 22.6 (63 306) |

Values are percentages (numbers) unless stated otherwise.
*Children with missing data were excluded (17% of the original cohort). Data on variables were available in 35%–100% of the excluded children.
†Gestational age categorised as extremely premature (22–27 weeks), very premature (28–31 weeks), moderate premature (32–36 weeks), term (37–41 weeks) and post-term (≥42 weeks).
‡ICD-10 codes: Q00–Q99.
§BMI categorised as underweight (BMI <18.5 kg/m$^2$), normal (BMI 18.5–24.9 kg/m$^2$), overweight (BMI 25.0–29.9 kg/m$^2$) and obese (BMI ≥30.0 kg/m$^2$).
BMI, body mass index; ICD-10, International Classification of Diseases, 10th revision; LGA, large for gestational age; SGA, small for gestational age.

childhood. Additionally, children born LGA or SGA were more likely to be hospitalised for sepsis and children exposed to maternal smoking during the pregnancy were more likely to be hospitalised for bacterial meningitis. Finally, the incidence of sepsis and bacterial meningitis decreased by the end of the study period.

### Incidence rates

The incidence rates of sepsis and bacterial meningitis hospitalisations in our study were similar to those reported in previous studies.[1 3 26] During the study period, the incidence rate of pneumococcal meningitis hospitalisations decreased substantially. This was expected, pneumococcal conjugate vaccines were introduced into the Swedish childhood immunisation programme between 2007 and 2009.[27] We have previously reported a decreasing incidence of pneumococcal meningitis after 2009 in Västerbotten county (Sweden).[26] No recent national study has examined the incidence of paediatric meningitis in Sweden.

### Perinatal factors

In our study, children born prematurely, SGA, with congenital malformations or with male sex were more likely to be hospitalised for sepsis and/or bacterial meningitis. These children are known to be more susceptible to infections in childhood. In a recent Australian cohort study, low gestational age was associated with an increased risk of infection-related deaths and hospital admissions throughout childhood, for example, gestational age <28 weeks was associated with 2.6 times higher risk of invasive bacterial infections.[12] A large Danish cohort study reported an association between low birth weight and infectious disease-related hospitalisations in the first 15 years of life, for example, children with a birth weight <1000 g were 3.3 times more likely to be hospitalised for sepsis.[13] Another Danish study found that congenital malformations and chromosomal abnormalities were associated with respiratory syncytial virus infections in the first 2 years of life.[28] Finally, previous studies have shown

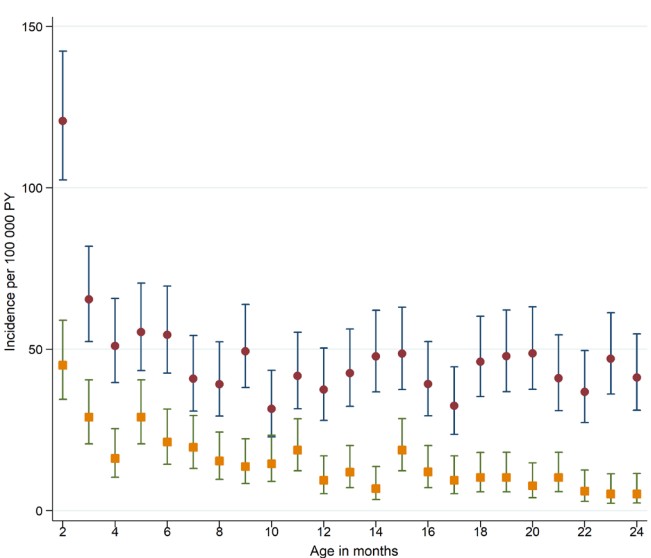

**Figure 2** Incidence of sepsis (brown circles) and bacterial meningitis (orange squares) in early childhood, by age in months. Incidence rates calculated as the number of hospital admissions per 100 000 person-years (PY) at risk. Vertical lines represent 95% CIs. Analyses included 1 406 547 children.

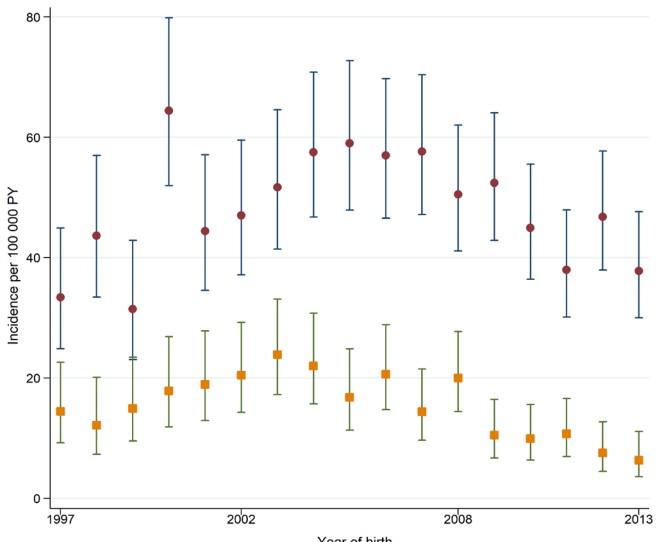

**Figure 3** Incidence of sepsis (brown circles) and bacterial meningitis (orange squares) in early childhood, by year of birth. Incidence rates calculated as the number of hospital admissions per 100 000 person-years (PY) at risk. Vertical lines represent 95% CIs. Analyses included 1 406 547 children.

**Table 2** Associations between perinatal factors and hospitalisations for sepsis and bacterial meningitis in early childhood

| | Sepsis | | Bacterial meningitis | |
|---|---|---|---|---|
| | Crude IRR (95% CI) | Adjusted IRR (95% CI) | Crude IRR (95% CI) | Adjusted IRR (95% CI) |
| **Sex** | | | | |
| Male | 1.12 (0.97 to 1.30) | 1.09 (0.94 to 1.26) | 1.35 (1.09 to 1.66) | 1.33 (1.08 to 1.64) |
| Female | 1 ref | 1 ref | 1 ref | 1 ref |
| **Gestational age*** | | | | |
| Extremely preterm | 15.73 (10.49 to 23.59) | 10.37 (6.78 to 15.86) | 7.50 (2.80 to 20.10) | 6.22 (2.28 to 16.94) |
| Very preterm | 6.44 (4.07 to 10.18) | 4.06 (2.49 to 6.63) | 5.99 (2.89 to 12.45) | 4.95 (2.39 to 10.24) |
| Moderate preterm | 2.17 (1.71 to 2.76) | 1.85 (1.45 to 2.37) | 1.41 (0.92 to 2.18) | 1.31 (0.84 to 2.03) |
| Term | 1 ref | 1 ref | 1 ref | 1 ref |
| Post-term | 0.80 (0.59 to 1.07) | 0.80 (0.59 to 1.08) | 0.93 (0.62 to 1.41) | 0.90 (0.59 to 1.36) |
| **SGA** | | | | |
| Yes | 3.51 (2.72 to 4.53) | 2.44 (1.83 to 3.25) | 1.89 (1.08 to 3.30) | 1.36 (0.78 to 2.40) |
| No | 1 ref | 1 ref | 1 ref | 1 ref |
| **LGA** | | | | |
| Yes | 1.69 (1.18 to 2.42) | 1.71 (1.19 to 2.44) | 1.43 (0.89 to 2.31) | 1.51 (0.93 to 2.45) |
| No | 1 ref | 1 ref | 1 ref | 1 ref |
| **Congenital malformation†** | | | | |
| Yes | 4.41 (3.61 to 5.40) | 3.89 (3.17 to 4.77) | 1.86 (1.21 to 2.88) | 1.69 (1.09 to 2.62) |
| No | 1 ref | 1 ref | 1 ref | 1 ref |
| **Maternal age** | | | | |
| Per year of age | 0.99 (0.98 to 1.01) | 0.99 (0.97 to 1.00) | 0.99 (0.96 to 1.01) | 0.99 (0.97 to 1.02) |
| Maternal smoking | | | | |
| Yes | 1.08 (0.85 to 1.38) | 0.96 (0.75 to 1.25) | 1.65 (1.22 to 2.24) | 1.46 (1.05 to 2.04) |
| No | 1 ref | 1 ref | 1 ref | 1 ref |
| **Pregnancy BMI‡** | | | | |
| Underweight | 0.84 (0.52 to 1.37) | 0.77 (0.48 to 1.26) | 0.95 (0.49 to 1.85) | 0.88 (0.45 to 1.71) |
| Normal | 1 ref | 1 ref | 1 ref | 1 ref |
| Overweight | 1.13 (0.95 to 1.33) | 1.10 (0.92 to 1.30) | 1.08 (0.84 to 1.39) | 1.06 (0.82 to 1.36) |
| Obese | 1.21 (0.96 to 1.52) | 1.11 (0.88 to 1.39) | 1.06 (0.77 to 1.46) | 0.99 (0.72 to 1.37) |
| **Parity** | | | | |
| Per child | 1.05 (0.98 to 1.13) | 1.08 (1.00 to 1.16) | 0.98 (0.87 to 1.10) | 0.96 (0.85 to 1.08) |
| **Maternal education in years** | | | | |
| ≤9 | 1.27 (1.01 to 1.59) | 1.07 (0.83 to 1.39) | 1.70 (1.23 to 2.36) | 1.39 (0.95 to 2.02) |
| 10–12 | 0.95 (0.80 to 1.13) | 0.89 (0.74 to 1.07) | 1.23 (0.96 to 1.59) | 1.09 (0.83 to 1.43) |
| 13–14 | 0.93 (0.73 to 1.18) | 0.90 (0.71 to 1.15) | 1.46 (1.06 to 2.02) | 1.37 (0.98 to 1.90) |
| ≥15 | 1 ref | 1 ref | 1 ref | 1 ref |
| **Maternal country of birth** | | | | |
| Sweden | 1 ref | 1 ref | 1 ref | 1 ref |
| Other Nordic | 1.03 (0.62 to 1.72) | 1.01 (0.61 to 1.69) | 1.23 (0.56 to 2.70) | 1.18 (0.54 to 2.58) |
| Other Europe and North America | 1.36 (1.07 to 1.72) | 1.33 (1.05 to 1.70) | 1.09 (0.73 to 1.64) | 1.10 (0.73 to 1.66) |
| Asia | 1.24 (0.96 to 1.61) | 1.17 (0.90 to 1.53) | 0.92 (0.60 to 1.42) | 0.94 (0.61 to 1.46) |
| Africa | 1.12 (0.65 to 1.94) | 1.00 (0.57 to 1.75) | 1.50 (0.84 to 2.65) | 1.60 (0.91 to 2.80) |
| Other | 0.88 (0.47 to 1.62) | 0.85 (0.46 to 1.57) | 1.37 (0.54 to 3.49) | 1.38 (0.54 to 3.52) |
| **Year of birth** | | | | |
| 1997–2002 | 1 ref | 1 ref | 1 ref | 1 ref |
| 2003–2008 | 1.26 (1.05 to 1.50) | 1.23 (1.03 to 1.48) | 1.18 (0.93 to 1.49) | 1.23 (0.96 to 1.57) |

Continued

**Table 2** Continued

| | Sepsis | | Bacterial meningitis | |
|---|---|---|---|---|
| | Crude IRR (95% CI) | Adjusted IRR (95% CI) | Crude IRR (95% CI) | Adjusted IRR (95% CI) |
| 2009–2013 | 1.00 (0.83 to 1.20) | 0.98 (0.80 to 1.19) | 0.54 (0.41 to 0.72) | 0.57 (0.42 to 0.77) |
| Observations (N) | 1 406 547 | 1 406 547 | 1 406 547 | 1 406 547 |
| Clusters (maternal ID) | 891 950 | 891 950 | 891 950 | 891 950 |

Analyses excluded children with missing data (n=280 202), leaving 1 406 547 children. Adjusted analyses were controlled for sex, gestational age, SGA, LGA, congenital malformation, maternal age, smoking during pregnancy, pregnancy BMI, parity, maternal education level, maternal country of birth and year of birth.
*Gestational age categorised as extremely premature (22–27 weeks), very premature (28–31 weeks), moderate premature (32–36 weeks), term (37–41 weeks) and post-term (≥42 weeks).
†ICD-10 codes: Q00–Q99.
‡BMI categorised as underweight (BMI <18.5 kg/m$^2$), normal (BMI 18.5–24.9 kg/m$^2$), overweight (BMI 25.0–29.9 kg/m$^2$) and obese (BMI ≥30.0 kg/m$^2$).
BMI, body mass index; ICD-10, International Classification of Diseases, 10th revision; IRR, incidence rate ratio; LGA, large for gestational age; SGA, small for gestational age.

that male sex is associated with a large number of childhood infections, including sepsis and meningitis.[3] [29] [30] Differences in sex hormones has been suggested as one possible explanation, testosterone has an overall suppressive effect on the immune system whereas oestrogen promotes Th1 cellular immune responses and humoral immunity.[29] Overall, these findings are consistent with our results.

In this study, children born LGA were more likely to be hospitalised for sepsis and bacterial meningitis. One explanation could be a higher incidence of maternal diabetes in children born LGA. Both maternal pre-existing diabetes and gestational diabetes are associated with an increased risk of childhood infections.[31] Similar to our results, a national cohort study reported an increased risk for sepsis in children with a birth weight >4500 g; however, in contrast to our results, the risk of bacterial meningitis was not increased in children born with high birth weight.[13] Therefore, our findings must be interpreted with caution.

We found an association between maternal smoking during pregnancy and bacterial meningitis. This was anticipated from previous research, children exposed to maternal smoking seems to be at risk of meningococcal infections. In a large cohort study from the UK, children exposed to maternal smoking during pregnancy were more likely to be hospitalised for meningococcal meningitis; no association was found for haemophilus meningitis.[32] Moreover, a US study reported that maternal smoking during pregnancy was associated with 2.9 times higher risk of invasive meningococcal disease in children 3 years and younger.[30] Maternal smoking during pregnancy is a proxy for maternal smoking after birth.[32] Cigarette smoke may increase the risk of bacterial meningitis

**Table 3** Interactions between sex, prematurity and perinatal factors

| | Sepsis | Bacterial meningitis |
|---|---|---|
| | RERI (95% CI) | RERI (95% CI) |
| RERI with sex* | | |
| Prematurity sex* | −0.78 (−1.78 to 0.22) | 1.14 (−0.24 to 2.53) |
| SGA sex* | −0.45 (−1.87 to 0.95) | 1.83 (−0.08 to 3.73) |
| LGA sex* | 0.04 (−1.20 to 1.27) | 1.11 (−0.49 to 2.71) |
| Congenital malformations sex* | −0.46 (−2.06 to 1.13) | −0.96 (−2.80 to 0.88) |
| Maternal smoking sex* | −0.56 (−1.37 to 0.25) | 0.07 (−0.61 to 0.75) |
| RERI with prematurity† | | |
| SGA prematurity* | 1.00 (−1.36 to 3.36) | 0.36 (−2.73 to 3.45) |
| LGA prematurity* | −0.97 (−3.01 to 1.08) | 1.58 (−2.76 to 5.91) |
| Congenital malformations prematurity* | 7.75 (3.45 to 12.06) | −0.33 (−3.19 to 2.53) |
| Maternal smoking prematurity* | −0.51 (−2.12 to 1.11) | −0.09 (−2.33 to 2.16) |

Analyses excluded children with missing data (n=280 202), leaving 1 406 547 children. Analyses were controlled for sex, gestational age, SGA, LGA, congenital malformation, maternal age, smoking during pregnancy, pregnancy BMI, parity, maternal education level, maternal country of birth and year of birth. Gestational age was included as a binary variable (0=gestational age≥37 weeks, 1=gestational age<37 weeks); other variables were included as previously described.
*Female is the reference category.
†Gestational age ≥37 weeks is the reference category.
BMI, body mass index; LGA, large for gestational age; RERI, relative excess risk due to interaction; SGA, small for gestational age.

by decreasing bronchial ciliary activity, decreasing neutrophil function and by predisposing children to viral respiratory infection.[30]

Numerous studies have shown that social deprivation is associated with an increased risk of disease including paediatric infections. In a Danish cohort study, low maternal education level was associated with 1.3 times higher risk of hospitalisation for infectious diseases in the first 5 years of life.[33] A national cohort study from New Zealand reported a higher incidence of infection-related hospitalisations in children belonging to economically deprived minorities.[34] Additionally, a US study reported that low maternal education was associated with a two times higher risk of invasive meningococcal disease in young children.[30] In our study, low maternal education was associated with an increased risk of hospitalisations for sepsis and bacterial meningitis in the crude analyses; however, these associations decreased and were non-significant in the adjusted analyses. This indicates that the effect of low maternal education was, at least partly, mediated by pregnancy and birth characteristics.

### Clinical implications

Pregnancy and birth characteristics are generally considered in clinical guidelines for neonatal infections.[9–11] In contrast, perinatal risk factors are not regularly considered in clinical guidelines for severe infections after the neonatal period, including Swedish and UK guidelines.[7 8 11 14] However, the Swedish guideline recommends that healthcare professionals consider chronic conditions including congenital malformations.[14] We found that prematurity, SGA, LGA, congenital malformations and maternal smoking during pregnancy were associated with severe infections in young children. Our findings are supported by previous studies.[12 13 28 30 32] Therefore, we suggest that clinical guidelines for risk assessment of infections after the neonatal period consider perinatal factors as well.

### Strengths and limitations

Strengths of this study include the use of high-quality national registers with information on a wide range of perinatal factors. This allowed us to systematically examine associations between perinatal factors and hospitalisations for severe paediatric infections. However, our study has several limitations. First, infections were identified using deidentified hospital discharge data. Consequently, cases could not be confirmed by laboratory findings or by other information obtained from medical records. Second, we were unable to examine categories of congenital malformations. This is problematic since susceptibility to infections varies between different congenital conditions.[28] Moreover, the observed interaction between prematurity and congenital malformations on the risk of sepsis may be due to more severe malformations in prematurely born children. Third, all children in Sweden have access to free healthcare and working parents are entitled to paid childcare leave.[35] Consequently, associations between sociodemographic characteristics and severe infections may not be generalised to other settings with less extensive welfare systems. Fourth, our study lacks information on postnatal factors associated with infectious diseases for example, acquired chronic diseases, vaccination status and breast feeding. Finally, data were missing for a large proportion (17%) of children. However, results from multiple imputation models were overall consistent with results from the complete case models, indicating no major selection bias.

## CONCLUSIONS

In our study, several perinatal factors, specifically prematurity, SGA, LGA, congenital malformations and maternal smoking were associated with an increased risk of sepsis and/or bacterial meningitis in children after the immediate neonatal period up to 2 years of age. Therefore, we suggest that clinical guidelines for risk assessment of infections in young children consider perinatal factors.

**Acknowledgements** The authors would like to thank Per E. Gustafsson, PhD at the Department of Epidemiology and Global Health, Umeå University, for reviewing and revising the manuscript and Johan Boström, MD at Department of Clinical Sciences, Pediatrics, Umeå University for help in creating the dataset. Håkan Sjöberg, MEc at Statistics Sweden, for help in compiling the dataset.

**Contributors** SV conceptualised and designed the study together with S-AS, carried out analyses and drafted the initial manuscript. UK contributed to the study design, reviewed and revised the manuscript. TL contributed to the study design, reviewed and revised the manuscript. S-AS conceptualised and designed the study together with SV, coordinated and supervised the database, reviewed and revised the manuscript.

**Funding** This study was supported by the Unit of Research, Development and Education, Östersund Hospital, Östersund, Sweden JLL-930202 (to SV); ALF Umeå University, Umeå, Sweden RV-933162 (to S-AS).

**Competing interests** None declared.

**Patient consent for publication** Not applicable.

**Ethics approval** The study was approved by the Regional Ethics Board in Umeå, reference number 2012-265-31M and 2017-399-32M.

**Provenance and peer review** Not commissioned; externally peer reviewed.

**Data availability statement** Data may be obtained from third parties and are not publicly available. We used deidentified register data obtained from third parties. It includes sensitive information and some access restrictions may apply. Interested researchers need to obtain data directly from the National Board of Health and Welfare in Sweden (socialstyrelsen@socialstyrelsen.se) and Statistics Sweden ( scb@scb.se). Children included in the study were identified in the Medical Birth Register, data on hospitalisations were obtained from the Swedish National Patient Register and data on deaths were obtained from the Cause of Death Register. All of these registers are maintained by the National Board of Health and Welfare in Sweden. Data on maternal education were obtained from the Longitudinal Integration Database for Health Insurance and Labour market Studies and data on migration were obtained from The Total Population Register, both registers are maintained by Statistics Sweden.

**ORCID iDs**
Samuel Videholm http://orcid.org/0000-0002-1468-5771
Sven-Arne Silfverdal http://orcid.org/0000-0002-3606-3797

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
