## [Reviewer comments · BMJ Open]

ARTICLE DETAILS

TITLE (PROVISIONAL)	Perinatal factors and hospitalisations for severe childhood infections: a population-based cohort study in Sweden
AUTHORS	Videholm, Samuel; Kostenniemi, Urban; Lind, Torbjörn; Silfverdal, Sven-Arne

VERSION 1 – REVIEW

REVIEWER	Hertting , Olof Karolinska Institute, Paediatric Infectious diseases
REVIEW RETURNED	17-Jun-2021

GENERAL COMMENTS	This is an interesting study with important clinical implications. Several questions need to be addressed however. Why was meningitis chosen among the different focal infections arising from bloodborne bacteria? Why was two years chosen as age limit in this study? What about children with the Swedish temporary personal number? Are they included? There is no discussion around the increased risk of invasive infections (especially for meningitis) in boys. This finding stands out and needs to be addressed in more detail. Is other tobacco use other than smoking included in the study? Maternal snuff use or electronic cigarettes? Immunization status would be helpful. Can this be obtained in your national registers? When stating that you excluded children with missing data (n = 280 202), what data is this referring to? Any data? A certain type of data? Is one missing data point sufficient for exclusion? I don't understand "Data on variables were available in 35%–100% of the excluded children." If data on variables were available in 100%. they would be included, or? Minor comments: Explain "(around 95%)" on page 6, line 29, "in a dose-dependent manner" on page 10, line 34, should perhaps be "age-dependent"?
--

REVIEWER	Fathima, Parveen Telethon Kids Institute, Wesfarmers Centre of Vaccines and Infectious Diseases
REVIEW RETURNED	01-Jul-2021

GENERAL COMMENTS	Using a population-based birth cohort of 1.4 million children, the authors have identified perinatal risk factors associated with hospitalisations for sepsis and bacterial meningitis in early childhood. My only major concern is that in their analyses, the authors have adjusted for a range of pregnancy, birth and socio-demographic characteristics including congenital malformations.
--

	However, comorbid conditions have not been included in the analysis. Comorbid conditions including lung disease, renal disease, cystic fibrosis, asplenia, diabetes etc have been identified as risk factors for meningitis and sepsis. The authors have access to all hospital admissions with a long follow up time and so, it would be easy to identify comorbid conditions (even if diagnosed later but present from birth or early childhood) and include in the models. Few things to note Background Para 1: It would be good to state the time period of the global incidence of sepsis. Para 2: Please specify the neonatal period – up to 28 days? Para 2: Need to add references for the ‘few previous studies’. Methods Patient and Public Involvement: What does ‘without the involvement’ mean? Is it without individual consent (as for linked de-identified data) or without having any consumer engagement? The cohort included ‘children who resided in Sweden at 28 days of age’ – how is this identified? Which dataset? It would be valuable to readers who are not familiar with the Swedish health register if the authors could include more information about each dataset. May be as an Appendix/supplementary table that could also detail variables obtained from each dataset. Outcomes: Were children diagnosed with sepsis mutually exclusive to those diagnosed with bacterial meningitis? Need to clarify this. Also, the way it currently reads – ‘we included hospital admission with a principal diagnosis of sepsis and bacterial meningitis’, looks like sepsis and bacterial meningitis were grouped together which is not the case. Outcomes: The ICD codes used for sepsis – looks like only codes relating to bacterial (infection-related?) sepsis were chosen (melioidosis and gonococcal sepsis has not been included – however, case numbers might be tiny and so might not make a difference). Could the authors please clarify that surgery- or instrument- (eg catheter, intubation) related sepsis was not included. Statistical analysis – Is there any particular reasoning behind the grouping of the years? Also, have there any changes to the diagnostic practices or ICD coding system during the time period that might have impacted the outcome? Incidence rates: The authors state ‘the incidence rate of sepsis decreased until the 7th month of life’. Looking at Figure 1, there is a significant drop in the 3rd month of life after which it seems to be stable (the point estimates do seem to drop but the CIs all overlap). Similarly, the authors state ‘the incidence rate of bacterial meningitis decreased steadily during the first year of life’ – this is not the case according to Figure 1. Following the 4th month of life, the point estimates seem to be higher until the 9th month. Discussion Can the authors postulate why large for gestational age children are more likely to be hospitalised for sepsis and/or meningitis?
--	--

	Third para – please clarify the low maternal education was associated in the crude analysis. Typos New Zealand has been spelt wrong. Abstract Would be good to include the statistical method used – for example, negative binomial regression was used to obtain crude and adjusted estimates?
--	--

VERSION 1 – AUTHOR RESPONSE

Reviewer Reports:

Reviewer: 1

Dr. Olof Hertting, Karolinska Institute

Comments to the Author:

This is an interesting study with important clinical implications. Several questions need to be addressed however.

Why was meningitis chosen among the different focal infections arising from bloodborne bacteria?

Reply: Bacterial meningitis was chosen for several reasons:

- Bacterial meningitis (together with sepsis) is the invasive infection with the highest risk of morbidity/sequel or even mortality.
- Bacterial meningitis hospitalisations are coded using agent-specific ICD-codes. This allowed us to examine the effect of introducing a pneumococcal vaccine into the Swedish childhood immunisation program between 2007-2009.
- Urban Johansson Kostenniemi and Sven-Arne Silfverdal (coauthors) have examined the incidence of sepsis and meningitis in Västerbotten county (Sweden) in several studies. This study complements their previous studies using national data.

Why was two years chosen as age limit in this study?

Reply: The first years of life is a critical period for severe infections. We choose the first two years because it allowed us to follow a substantial cohort of children born before and after a pneumococcal vaccine was added to the Swedish childhood immunisation program 2007-2009. (our dataset contains information on hospitalisations until 2015)

What about children with the Swedish temporary personal number? Are they included?

Reply: The Medical Birth Register uses a personal identification number (PIN) for identification. Immigrants or refugees who had not yet received a Swedish personal identification number will not be included since linking to other registers is not possible. Immigrants will receive a PIN if they intend to stay in Sweden for at least 1 year. Immigrants who do not fulfil this criterion, but who e.g. are taxed in Sweden or who use the Swedish social security system will be assigned a coordination number. Individuals with coordination numbers are not registered in the national health registers. According to "The Swedish Medical Birth Register - A Summary of Content and Quality" linking of around 0,5% of all birth were not possible due to immigrants lacking PINs.

There is no discussion around the increased risk of invasive infections (especially for meningitis) in boys. This finding stands out and needs to be addressed in more detail.

Reply: The text has been revised:

"Finally, previous studies have shown that male sex is associated with a large number of childhood infections, including sepsis and meningitis.^{3 29 30} Differences in sex hormones has been suggested

as one possible explanation, testosterone has an overall suppressive effect on the immune system whereas estrogen promotes Th1 cellular immune responses and humoral immunity.²⁹

Is other tobacco use other than smoking included in the study? Maternal snuff use or electronic cigarettes?

Reply: Information on maternal snuff is available in the Medical Birth Register. However, only around 3% of mothers reported that they used snuff. Moreover, according to "The Swedish Medical Birth Register - A Summary of Content and Quality" the validity of this variable is questionable. We have no information on electronic cigarettes.

Immunization status would be helpful. Can this be obtained in your national registers?

Reply: We do not have individual-level data on vaccination for this cohort. This would be preferable and would be possible by linking data from the national vaccination register.

The text has been revised:

"Fourth, our study lacks information on postnatal factors associated with infectious diseases e.g. acquired chronic diseases, vaccination status and breastfeeding."

When stating that you excluded children with missing data (n = 280 202), what data is this referring to? Any data? A certain type of data? Is one missing data point sufficient for exclusion? I don't understand "Data on variables were available in 35%–100% of the excluded children." If data on variables were available in 100%. they would be included, or?

Reply: We conducted "Complete case analysis (listwise deletion)" meaning that only children with complete data on all variables were included (280 202 children had missing data in one or more variables). This method (in combination with multiple imputation) is often preferred over "Available case analysis (pairwise deletion)" since the same cases/children are included in the crude and adjusted analysis.

Table 1 presents information variables. Some variables had no missing data e.g. data on year of birth was complete. Consequently, information on year of birth was available for all excluded children. To be more clear, we have added a flowchart of the study population (figure 1).

Minor comments: Explain "(around 95%)" on page 6, line 29, "in a dose-dependent manner" on page 10, line 34, should perhaps be "age-dependent"?

Reply: The text has been revised: (performed in around 95% of pregnancies)

Reply: We choose the term "dose-dependent manner" to point out that the age shows a "dose-response relationship" with hospitalisations.

Reviewer: 2

Dr. Parveen Fathima, Telethon Kids Institute

Comments to the Author:

Using a population-based birth cohort of 1.4 million children, the authors have identified perinatal risk factors associated with hospitalisations for sepsis and bacterial meningitis in early childhood. My only major concern is that in their analyses, the authors have adjusted for a range of pregnancy, birth and socio-demographic characteristics including congenital malformations. However, comorbid conditions have not been included in the analysis. Comorbid conditions including lung disease, renal disease, cystic fibrosis, asplenia, diabetes etc have been identified as risk factors for meningitis and sepsis. The authors have access to all hospital admissions with a long follow up time and so, it would be easy to identify comorbid conditions (even if diagnosed later but present from birth or early childhood) and include in the models.

Reply: We agree that comorbid conditions are important risk factors for sepsis and bacterial meningitis. The main reason why we have not included comorbid conditions is because of lack of data. The National Inpatient Register includes information on hospitalisations for all causes. However, our dataset is restricted to infection and inflammatory diseases. The Swedish National Board of Health usually restricts the data they give out to protect personal integrity i.e. to avoid "backtracing". It

is also important to point out that (acquired) comorbid conditions occurs after pregnancy/birth and cannot be confounders to pregnancy or birth factors. Finally, examining the association between comorbid conditions and infections using register data is challenging due to the risk of “reverse causation”, many comorbid conditions will become symptomatic and be diagnosed during infection episodes. There is a risk that infection leads to a (diagnosed) comorbid condition and not the other way around.

Reply: The text has been revised:

“Fourth, our study lacks information on postnatal factors associated with infectious diseases e.g. acquired chronic diseases, vaccination status and breastfeeding”

Few things to note

Background

Para 1: It would be good to state the time period of the global incidence of sepsis.

Reply: The text has been revised: “in 2013” “in 2016”

Para 2: Please specify the neonatal period – up to 28 days?

Reply: The text has been revised: “(from birth until 28 days)”

Para 2: Need to add references for the ‘few previous studies’.

Reply: This statement was based on our literature review, we have removed the sentence.

Methods

Patient and Public Involvement: What does ‘without the involvement’ mean? Is it without individual consent (as for linked de-identified data) or without having any consumer engagement?

Reply: We mean that patients/the public were not involved in the design or conduct of our study. Neither was Individual consent obtained, but this is stated under “Patient consent”.

The text has been revised:” This study used retrospectively collected register data. Therefore, patients or members of the public were not involved in the design or conduct of this study.”

The cohort included ‘children who resided in Sweden at 28 days of age’ – how is this identified? Which dataset?

Reply: We agree this was unclear. The text has been revised:” The Medical Birth Register included 1 690 100 live births between 1997 and 2013. Children who died (n = 3193) or emigrated (n = 158) in the neonatal period were removed, leaving 1 686 749 children who resided in Sweden at 28 days of age (target population). We excluded children with missing data (n = 280 202), leaving 1 406 547 children (figure 1).”

Morover, a flowchart of the study population (figure 1) have been added.

It would be valuable to readers who are not familiar with the Swedish health register if the authors could include more information about each dataset. May be as an Appendix/supplementary table that could also detail variables obtained from each dataset.

Reply: We have added some information about each register. A detailed description of each register is provided in references 15-18. The text has been revised: “The dataset included information from the Medical Birth Register which covers over 98-99% of all births and contains information on prenatal, delivery and neonatal care, the National Inpatient Register which holds information on 99% of all inpatient hospital admissions, the Longitudinal integration database for health, insurance and labour market studies which contains socio-economic data, the Cause of Death Register which covers over 97% of all deaths and the Total Population Register.15-18”

This study uses a large and complex cohort that was created by combining several national and local registers. The cohort is too complex/contains too many variables to be described in an appendix. Nevertheless, we understand that a description of the cohort would be beneficial e.g. presented as a “cohort profile”.

Outcomes: Were children diagnosed with sepsis mutually exclusive to those diagnosed with bacterial meningitis? Need to clarify this. Also, the way it currently reads – ‘we included hospital admission with a principal diagnosis of sepsis and bacterial meningitis’, looks like sepsis and bacterial meningitis were grouped together which is not the case.

Reply: The text has been revised:” During the study period, 1011 children were hospitalised at least once for sepsis, 382 children were hospitalised at least once for bacterial meningitis and 18 children were hospitalised at least once for both sepsis and bacterial meningitis.”

“The two main outcomes were number of hospitalisations for sepsis and number of hospitalisations for bacterial meningitis. Information on hospital admissions with a principal diagnosis of sepsis (ICD-10 codes: A39.1-A39.9, A40, A41, A48.3, A02.1 and A32.7) and bacterial meningitis (ICD-10 codes: A39.0, A32.1 and G00) were retrieved from the National Inpatient Register. We included only hospitalisations with admission dates between 28 days and 2 years of age. Consequently, infections with admission date in the neonatal period were not included. Readmissions within 30 days were excluded. . “

Outcomes: The ICD codes used for sepsis – looks like only codes relating to bacterial (infection-related?) sepsis were chosen (melioidosis and gonococcal sepsis has not been included – however, case numbers might be tiny and so might not make a difference). Could the authors please clarify that surgery- or instrument- (eg catheter, intubation) related sepsis was not included.

Reply: Under the assumption that clinicians used the right ICD-10 code, surgery- or instrument sepsis will not be included. However, this can not be confirmed in our data. The text has been revised:

“Consequently, cases could not be confirmed by laboratory findings or by other information obtained from medical records.”

Statistical analysis – Is there any particular reasoning behind the grouping of the years? Also, have there any changes to the diagnostic practices or ICD coding system during the time period that might have impacted the outcome?

Reply: A pneumococcal vaccine was added to the Swedish childhood immunisation program 2007-2009. We grouped year of birth so we got a substantial cohort of vaccinated and unvaccinated children. Children born 1997–2002 were not vaccinated, some children born 2003–2008 were vaccinated (depending on region) and all children born 2009–2013 were vaccinated (if they followed the vaccination program). We used the same year grouping in the regression analyses and when we estimated meningitis incidence rates. The incidence of pneumococcal meningitis was 8.3 per 100 000 PY in 1997–2002, 10.0 per 100 000 PY in 2003–2008 and 2.2 per 100 000 PY in 2009–2013. (Appendix A).

No major changes in the ICD coding system occurred during the study period. ICD-10 was introduced 1997. National recommendations for coding of sepsis according to SOFA-score was introduced in 2020 (after the study period ended).

Incidence rates: The authors state ‘the incidence rate of sepsis decreased until the 7th month of life’. Looking at Figure 1, there is a significant drop in the 3rd month of life after which it seems to be stable (the point estimates do seem to drop but the CIs all overlap). Similarly, the authors state ‘the incidence rate of bacterial meningitis decreased steadily during the first year of life’ – this is not the case according to Figure 1. Following the 4th month of life, the point estimates seem to be higher until the 9th month.

Reply: The text has been revised: “The incidence rate of sepsis decreased until the 3rd month of life. Thereafter, it remained around 45 hospitalisations per 100 000 PY. The incidence rate of bacterial meningitis decreased until the 3rd month of life, thereafter it was around 20 hospitalisations per 100 000 PY in the first year of life and around 10 hospitalisations per 100 000 PY in the second year of life.”

Discussion

Can the authors postulate why large for gestational age children are more likely to be hospitalised for sepsis and/or meningitis?

Reply: The association between large for gestational age and severe infections is puzzling. One explanation may be an increased risk due to a higher prevalence of maternal diabetes.

The text has been revised: "In this study, children born large for gestational age were more likely to be hospitalised for sepsis and bacterial meningitis. One explanation could be a higher incidence of maternal diabetes in children born large for gestational age. Both maternal pre-existing diabetes and gestational diabetes are associated with an increased risk of childhood infections.³¹ Similar to our results, a national cohort study reported an increased risk for sepsis in children with a birth weight >4500; however, in contrast to our results, the risk of bacterial meningitis was not increased in heavy children.¹³ Therefore, our findings must be interpreted with caution."

Third para – please clarify the low maternal education was associated in the crude analysis.

Reply: The text has been revised: "In this study, low maternal education was associated with an increased risk of sepsis and bacterial meningitis in the crude analyses."

Typos

New Zealand has been spelt wrong.

Reply: The text has been revised: "New Zealand"

Abstract

Would be good to include the statistical method used – for example, negative binomial regression was used to obtain crude and adjusted estimates?

Reply: The text has been revised: "Associations between perinatal factors and hospitalisations were examined using negative binomial regression models. "

VERSION 2 – REVIEW

REVIEWER	Hertting , Olof Karolinska Institute, Paediatric Infectious diseases
REVIEW RETURNED	30-Aug-2021

GENERAL COMMENTS	Thank you for responding to the comments I had.
---

REVIEWER	Fathima, Parveen Telethon Kids Institute, Wesfarmers Centre of Vaccines and Infectious Diseases
REVIEW RETURNED	19-Aug-2021

GENERAL COMMENTS	I'm happy with the authors' responses to the reviewers' comment and the changes made to the manuscript. Just a few minor suggestions/edits: Background Para 1: Typos and tense – "the global incidence of paediatric sepsis was estimated to be 1.2 million cases per year in 2013 and the global incidence of pediatric meningitis was around 280 000 cases per year". Methods Outcomes: Just to make it clear, suggest using 'or' – "with a principal diagnosis of sepsis (ICD-10 codes: A39.1-A39.9, A40, A41, A48.3, A02.1 and A32.7) and/or bacterial meningitis (ICD-10 codes: A39.0, A32.1 and G00)" Incidence rates: For bacterial meningitis it should be "decreased
---

	until the 4th months of life” – not 3rd. Discussion Please avoid the use of the term ‘heavy’ children. This is just a suggestion (happy for the authors to ignore): With respect to the low maternal education, could consider revising the paragraph as follows: “Numerous studies have shown that social deprivation is associated with an increased risk of disease including paediatric infections. In a Danish cohort study, low maternal education level was associated with 1.3 times higher risk of hospitalisation for infectious diseases in the first 5 years of life.³¹ A national cohort study from New Zealand reported a higher incidence of infection-related hospitalisations in children belonging to economically deprived minorities i.e. the Māori and Pacific peoples.³² Additionally, a US study reported that low maternal education was associated with a two times higher risk of invasive meningococcal disease in young children.³⁰ In our study, low maternal education was associated with an increased risk of hospitalisations for sepsis and bacterial in the crude analyses; however, these associations decreased and were non-significant in the adjusted analyses. This indicates that the effect of low maternal education was, at least partly, mediated by pregnancy and birth characteristics.”
--	--

VERSION 2 – AUTHOR RESPONSE

Response to reviewers:

Reviewer: 1

Dr. Olof Hertting , Karolinska Institute

Comments to the Author:

Thank you for responding to the comments I had.

Reviewer: 2

Dr. Parveen Fathima, Telethon Kids Institute

Comments to the Author:

I'm happy with the authors' responses to the reviewers' comment and the changes made to the manuscript.

Just a few minor suggestions/edits:

Background

Para 1: Typos and tense – “the global incidence of paediatric sepsis was estimated to be 1.2 million cases per year in 2013 and the global incidence of pediatric meningitis was around 280 000 cases per year”.

Reply: The text has been revised as suggested.

Methods

Outcomes: Just to make it clear, suggest using ‘or’ – “with a principal diagnosis of sepsis (ICD-10 codes: A39.1-A39.9, A40, A41, A48.3, A02.1 and A32.7) and/or bacterial meningitis (ICD-10 codes: A39.0, A32.1 and G00)”

Reply: The text has been revised as suggested.

Incidence rates: For bacterial meningitis it should be “decreased until the 4th months of life” – not 3rd.

Reply: The text has been revised as suggested.

Discussion

Please avoid the use of the term ‘heavy’ children.

Reply: The text has been revised as suggested.

This is just a suggestion (happy for the authors to ignore): With respect to the low maternal education, could consider revising the paragraph as follows:

“Numerous studies have shown that social deprivation is associated with an increased risk of disease including paediatric infections. In a Danish cohort study, low maternal education level was associated with 1.3 times higher risk of hospitalisation for infectious diseases in the first 5 years of life.³¹ A national cohort study from New Zealand reported a higher incidence of infection-related hospitalisations in children belonging to economically deprived minorities i.e. the Māori and Pacific peoples.³² Additionally, a US study reported that low maternal education was associated with a two times higher risk of invasive meningococcal disease in young children.³⁰ In our study, low maternal education was associated with an increased risk of hospitalisations for sepsis and bacterial in the crude analyses; however, these associations decreased and were non-significant in the adjusted analyses. This indicates that the effect of low maternal education was, at least partly, mediated by pregnancy and birth characteristics.”

Reply: The text has been revised as suggested.